# Neuroprotective Effect of Red Sea Marine Sponge *Xestospongia testudinaria* Extract Using In Vitro and In Vivo Diabetic Peripheral Neuropathy Models

**DOI:** 10.3390/ph15111309

**Published:** 2022-10-24

**Authors:** Rania Magadmi, Kariman Borouk, Diaa T. A. Youssef, Lamiaa A. Shaala, Aziza R. Alrafiah, Rasheed A. Shaik, Sameer E. Alharthi

**Affiliations:** 1Pharmacology Department, Faculty of Medicine, King Abdulaziz University, Jeddah 22254, Saudi Arabia; 2Department of Natural Products, Faculty of Pharmacy, King Abdulaziz University, Jeddah 21589, Saudi Arabia; 3Natural Products Unit, King Fahd Medical Research Center, King Abdulaziz University, Jeddah 21589, Saudi Arabia; 4Suez Canal University Hospital, Suez Canal University, Ismailia 41522, Egypt; 5Department of Medical Laboratory Technology, Faculty of Applied Medical Sciences, King Abdulaziz University, Jeddah 21589, Saudi Arabia; 6Department of Pharmacology & Toxicology, Faculty of Pharmacy, King Abdulaziz University, Jeddah 21589, Saudi Arabia

**Keywords:** Red Sea marine sponge, *Xestospongia testudinaria*, antioxidant, anti-inflammatory, diabetic peripheral neuropathy

## Abstract

Diabetic peripheral neuropathy (DPN) is a common complication of diabetes. Oxidative stress plays an important role in the pathophysiology of DPN. Red Sea marine sponge *Xestospongia testudinaria* extract has a promising neuroprotective effect, presumably owing to its antioxidant and anti-inflammatory properties. Thus, this study aimed to investigate the neuroprotective effect of the sponge *X. testudinaria* extract on in vitro and in vivo models of DPN. Mice dorsal root ganglia (DRG) were cultured with high glucose (HG) media and used as an in vitro model of DPN. Some of the DRGs were pre-treated with 2 mg/mL of *X. testudinaria*. The *X. testudinaria* extract significantly improved the HG-induced decreased neuronal viability and the neurite length. It improved the oxidative stress biomarkers in DRG cultures. The DPN model was induced in vivo by an injection of streptozotocin at a dose of 150 mg/kg in mice. After 35 days, 0.75 mg/kg of the *X. testudinaria* extract improved the hot hyperalgesia and the DRG histology. Although the sponge extract did not reduce hyperglycemia, it ameliorated the oxidative stress markers and pro-inflammatory markers in the DRG. In conclusion, the current study demonstrates the neuroprotective effect of Red Sea sponge *X. testudinaria* extract against experimentally induced DPN through its antioxidant and anti-inflammatory mechanisms.

## 1. Introduction

Diabetic peripheral neuropathy (DPN) is one of the most common complications of diabetes (Boulton, 2005) and it affects 50% of both type 1 and type 2 patients [1]. A recent multicentric survey conducted in Saudi Arabia showed that a third of diabetic patients had a painful diabetic peripheral neuropathy [1]. In the US, the health costs associated with DPN per year are approximately 4.6–13.7 billion USD, equal to a quarter of the total expenditure per year on DM [2].

Although many therapeutic approaches have become available to relieve diabetic neuropathy symptoms [3], treatments remain somewhat unsatisfactory. However, recent research has uncovered significant knowledge on the pathogenesis of diabetic neuropathy. It has been suggested that hyperglycemia triggers oxidative stress, which plays an important role in neurotoxicity through the over-production of reactive oxygen species [3,4,5,6]. This leads to mitochondrial dysfunction [7], neuronal damage [8], and finally apoptosis [9].

*Xestospongia testudinaria (X. testudinaria*) (Family: Petrosiidae), known as barrel sponges, represent a large group of marine sponges found in Red Sea coral reefs. Members of the genus *Xestospongia* are among the richest resources of pharmacologically active compounds isolated from marine organisms [10,11]. Recently, some studies have shown the cytotoxic, anti-inflammatory, and antioxidant effects of *X. testudinaria* [11,12,13,14]. Gas chromatography-mass spectroscopy (GC-MS) analysis has revealed that *X. testudinaria* extract contains 41 significant compounds, such as: glutamine, pyrimidine, and hexadecenoic acid [12]. This study investigates the potential neuroprotective effects of Red Sea *X. testudinaria* on diabetic peripheral neuropathy in in vitro and in vivo models.

## 2. Results

### 2.1. Effect of Sponge Extract on In Vitro Model of Diabetic Neuropathy

#### 2.1.1. High Glucose (HG)-Induced Decreases in DRG Neuronal Viability and Neurite Length

To evaluate the neuroprotective effect of diabetic neuropathy, DRG cultures with high glucose (HG) media were pre-incubated with various concentrations of *X. testudinaria* extract (0.5, 1, 2 mg/mL). The HG media induced a significant reduction in neuronal viability (Figure 1A) and neurite length (Figure 1B,C) in DRG cultures of 26% and 70%, respectively, compared to the control. However, a pre-incubation with the sponge extract significantly increased in the DRGs viability and significantly ameliorated the HG effect on the neuronal length. Interestingly, the extract of *X. testudinaria* demonstrated a neuroprotective effect higher than α-lipoic acid.

#### 2.1.2. Effect of *X. testudinaria* Extract on HG-Induced Oxidative Stress in DRG Culture

The incubation of neurons with a HG media caused more than a 3-fold increase in malondialdehyde (MDA) concentration compared to the control group. However, a pre-treatment with 2 mg/mL of *X. testudinaria* extract ameliorated the elevation of the MDA concentration, causing a significant reduction in about 60% from the HG group, without any significant change from the control, as shown in (Figure 2A). Similarly, the treatment with α-lipoic acid significantly reduced the MDA concentration by about 50% compared to the HG group.

By contrast, the HG media significantly depleted the glutathione (GSH) content and superoxide dismutase enzymes (SOD) activity in the DRG culture by about 60% and 50%, respectively, from the control value (Figure 2B,C). However, treating the cultures with 2 mg/mL of *X. testudinaria* extract significantly prevented GSH and SOD depletion. It is noteworthy that the *X. testudinaria* extract was able to restore the GSH content to the value of the control group. Likewise, the effect of the α-lipoic acid was significant in compensating the depleted GSH content and SOD activity to the control group levels.

### 2.2. Evaluation of Acute Toxicity of X. testudinaria Extract in Mice

The LD50 study showed no mortality reported or any toxicity signs after 24 h of intraperitoneal (IP) injection of 2000 mg/kg of *X. testudinaria* extract. Moreover, mice in the 2000 mg/kg satellite group showed normal behavior during the 14 days after the IP injection.

For acute toxicity, the IP administration of 0.75 mg/kg of the sponge extract showed the normal biochemical and histological features of the liver and pancreas, as shown in Appendix A.

### 2.3. Effect of X. testudinaria Extract on In Vivo Model of Diabetic Neuropathy

Based on the toxicology results, 0.75 mg/kg of *X. testudinaria* extract was selected for the subsequent experiments using the streptozotocin (STZ)-induced diabetic neuropathy model in mice.

#### 2.3.1. Effect of *X. testudinaria* Extract on STZ-Induced Hot Hyperalgesia

As shown in Figure 3, there was a significant increase in the paw withdrawal latency to hot stimuli (hyperalgesia) in the diabetic group at the end of the experiment compared to before the STZ administration. The administration of 0.75 mg/kg of *X. testudinaria* extract induced a significant rise of the withdrawal latency to hot stimuli, compared to the diabetic group. Interestingly, the administration of *X. testudinaria* extract alone did not affect the mice response to hot stimuli.

#### 2.3.2. Effect of *X. testudinaria* Extract on DRG Histology

The examination of the H- and E-stained sections of the STZ-diabetic group showed an apparent reduction in the number of ganglion cells. The satellite cell numbers had increased surrounding the neurons (Figure 4C). The sections of the STZ-diabetic with *X. testudinaria*-treated group (Figure 4D) appeared with a normal distribution of the different size neurons and the satellite cells were arranged in a single layer around the neurons, simulating the control section (Figure 4A) and sections from the *X. testudinaria*-only group (Figure 4B).

The examination of the reticulin-stained sections of the treated diabetic group with *X. testudinaria* extract (Figure 5D) showed multiple rounded, large, medium, and small neurons, and there were thinly dispersed and uniform single-cell layers of satellite cells around the cell bodies. Those morphological characteristics are similar to the control group (Figure 5A) and the *X. testudinaria*-only group (Figure 5B). In the STZ-diabetic group, the sections of the DRG showed an apparent reduction in the number of ganglion cells. The satellite cells around the neurons appeared as multilayers (Figure 5C).

#### 2.3.3. Effect of *X. testudinaria* Extract on Glucose Level

As shown in Figure 6, the administration of STZ significantly increased the FBG in the STZ- and *X. testudinaria*-treated groups compared to the control group. There was a significant increase in the FBS in week 5, by 17%, for the STZ group compared to week 1. Remarkably, there was no significant difference in the FBG between the administrations of the *X. testudinaria* extract compared to the control group.

#### 2.3.4. Effect of *X. testudinaria* Extract on STZ-Induced Oxidative Stress in a Mice Model

The DRG of the STZ-diabetic neuropathy mice showed a significant increase in the MDA concentration, one greater than 4-fold of the control (Figure 7A). However, the treatment with *X. testudinaria* extract significantly decreases the concentration of MDA; a 45% decrease compared to the STZ group. Similarly, STZ-induced diabetic neuropathy significantly depleted the content of GSH and SOD by about 90% and 58%, respectively, compared to the control value (Figure 7B,C). By contrast, animals treated with the *X. testudinaria* extract significantly ameliorated the GSH and SOD depletion caused by STZ; a 7-fold decrease compared to the STZ group. Remarkably, there was no significant difference in the content of MDA, GSH, and SOD in the DRG of the group that received the *X. testudinaria* extract compared to the control.

#### 2.3.5. Effect of *X. testudinaria* Extract on Pro-Inflammatory Biomarkers in DRG of Diabetic Neuropathy Mice (TNF-α, NF-κB)

The pro-inflammatory tumor necrosis factor (TNF)-α and nuclear factor kappa B (NF-κB) content in the DRG of the STZ group significantly increased to approximately >20% in the TNF-α and >70% in NF-κB compared with the control (Figure 8A,B). In contrast, a treatment with the *X. testudinaria* extract was able to significantly decrease the content of TNF-α by 27% and NF-κB by 64% compared with the STZ group. However, an administration of the *X. testudinaria* extract alone did not affect the content of the TNF-α and NF-κB in the DRG.

## 3. Discussion

Diabetic peripheral neuropathy is presented as spontaneous pain and hyperalgesia due to hyperglycemia-induced sensory neuronal dysfunction [15,16]. A large and growing body of literature has indicated that hyperglycemia-induced oxidative stress and inflammation are key pathways in developing diabetic neuropathy [4]. Thus, exploring natural therapeutic agents with antioxidant and anti-inflammatory activities is an attractive strategy to prevent diabetic peripheral neuropathy.

*X. testudinaria* is a member of the genus Xestospongia and is considered a rich source of natural antioxidants and anti-inflammatory compounds [11,14]. This study was carried out to investigate the potential neuroprotective effect of Red Sea *X. testudinaria* extract on diabetic peripheral neuropathy in both in vitro and in vivo models.

Several mechanisms have been identified that may explain hyperglycemia-induced neuropathy, such as oxidative stress [4]. An oxidative stress contribution could be either through the increase in the ROS generation or the decrease in the endogenous antioxidants [17]. The increase in ROS leads to the release of the pro-apoptotic factors, leading to the altering of the morphology of the neurons and, subsequently, cell death [17]. Indeed, the results of the current study support this mechanism. The incubation of DRG cultures with HG media caused a significant depletion of the endogenous antioxidants (GSH and SOD) and dramatically increased the lipid peroxide (MDA). However, the administration of 2 mg/mL of *X. testudinaria* extract showed a significant improvement in the neuronal viability and morphology. These effects could be explained by its antioxidant properties, shown by its ability to ameliorate the elevation of MDA and restore the depletion of the GSH and SOD compared with the HG group. To the best of our knowledge, this is the first in vitro experiment to demonstrate the neuroprotective effect of *X. testudinaria* extract in a diabetic peripheral neuropathy model.

Several previous studies support that hyperglycemia-induced oxidative stress is a key factor in inducing sensory neuron injury that manifests clinically as hyperalgesia [18]. The in vitro results from the current study suggest that *X. testudinaria* reverses this mechanism. Thus, to confirm this assumption, an in vivo animal model of diabetic peripheral neuropathy was required to validate the in vitro results.

Owing to a lack of detailed evidence regarding the safety of an IP administration of *X. testudinaria*, the acute toxicity study for the *X. testudinaria* extract was performed as an initial step to assess the harmful effect that can occur shortly following the administration of a new natural product or chemical [19]. Based on the toxicology results of the current study, the highest, non-hepatic toxic dose (0.75 mg/kg) was selected for the subsequent experiments, using the STZ-induced diabetic neuropathy mice model.

In the current study, the administration of 0.75 mg/kg of *X. testudinaria* extract significantly prevented hyperalgesia in diabetic mice. It prolongs the withdrawal latency to hot stimuli compared to the diabetic group. Hyperalgesia is mediated mainly by the C-fibers in DRG [20]. Thus, DRG histopathology was then carried out to confirm the neuroprotective effect of the *X. testudinaria* extract at the cellular level.

As the *X. testudinaria* extract was effective in behavioral experiments, the histopathology of DRG confirmed its neuroprotective effect. The administration of the *X. testudinaria* extract increased the number of ganglion cells compared to the STZ-group.

In the current study, the *X. testudinaria* extract did not affect the STZ-induced hyperglycemia. Thus, the neuroprotective effect of the *X. testudinaria* extract on diabetic neuropathy seen in the behavioral experiment is not mediated by the direct effect of the *X. testudinaria* extract on hyperglycemia. This could be due to the antioxidant effect of the *X. testudinaria* extract that has been reported in other studies [12,21].

Oxidative stress occurs when the quantity of reactive oxygen species (ROS) overcomes the quantity of neutralizing agents, termed endogenous antioxidants. Numerous in vitro and in vivo studies show that an increase in the glucose level can lead to the overproduction of oxidative stress biomarkers, such as MDA, and inhibits the synthesis of endogenous antioxidants [22,23]. Oxidative stress may contribute to vascular dysfunction in diabetes, owing to decreased nitric oxide (NO) [24] and vascular dysfunction further contributes to diabetic neuropathy pathogenesis. Furthermore, the excessive production of ROS leads to the destruction of the neuronal cell membrane, cell protein, and nucleic acid, which can eventually cause neuronal cell death [23]. Additionally, ROS interact with a lipid of the myelin sheath in neuron cells, resulting in the destruction of the axon of the nerve [25]. In a previous experiment on animals, more than 50% of large myelinated fiber loss was due to the overproduction of ROS [26]. This link represents major factors that contribute to the pathogenesis of diabetic micro-vascular and macro-vascular complications, including diabetic neuropathy [25]. Therefore, the oxidative stress is a possible mechanism of action of the neuroprotective effect of *X. testudinaria* extract in diabetic neuropathy.

The effect of *X. testudinaria* extract on STZ-induced oxidative stress was investigated in the current study. The administration of the *X. testudinaria* extract showed antioxidant properties manifested by the significantly improved GSH and SOD depletion in DRG caused by STZ. Likewise, *X. testudinaria* ameliorated the elevation of the MDA concentration. Collectively, *X. testudinaria* mediates an antioxidant defense system in the DRG of the diabetic mice. These results are consistent with findings from the current in vivo study.

A previous in vitro study has shown that Red Sea *X. testudinaria* extract exhibited antioxidant properties against DPPH free radicals [12]. The same study revealed the antioxidant activity of the *X. testudinaria* extract in vivo. The injection of 100 mg/kg of *X. testudinaria* in paw rat resulted in a significantly decreased paw MDA and NO and significantly increased the paw GSH content.

Previous studies have shown that the Australian *X. testudinaria* has caused vasodilation in the coronary and femoral arteries of anesthetized dogs [27], as well as resulting in isolated mesenteric arteries in rats [28]. Although the effect of *X. testudinaria* on blood vessels has not been investigated in the current study, this is another possible mechanism that could explain its antioxidant role, which may be responsible for the improvement in the hyperalgesia test and DRG histology of diabetic mice.

Pro-inflammatory mediators play a crucial role to inducing insulin resistance via the involvement of oxidative stress and the activation of various transcriptional mediated molecular or metabolic pathways [29]. The current study demonstrated the ability of *X. testudinaria* extract to reverse the abnormal elevation of TNF-α in the DRG compared to the DM group. TNF-α is one of the most important pro-inflammatory mediators that is critically involved in the development of diabetic peripheral neuropathy. It is reported that TNF-α suppresses the neurotrophic factors and induces a neuronal degeneration. It induces tissue-specific inflammation through the involvement of the generation of ROS and the activation of various transcriptional mediated pathways [30].

The role of *X. testudinaria* extract is in agreement with a previous study that showed that the pre-treatment of carrageenan-treated rats in vivo with 100 mg/kg of *X. testudinaria* extract reduced the local tissue elevated by TNF-α, IL-6, and IL-1β [12]. A further in vivo study carried out on a different barrel marine sponge obtained from *Aplysina caissara*, *Haliclona* sp., and *Dragmacidon reticulatum* showed that the oral administration of *A. cassara*, *Haliclona sp*. and *D. reticulatum* extracts significantly reduced the formalin-induced paw edema in mice. Taken together, these data show that marine sponges can be important sources of anti-inflammatory agents. However, the molecular mechanism of marine sponges’ anti-inflammatory properties has not yet been properly uncovered.

The NF-κB family is comprised of DNA-binding protein factors that are required for the transcription of many pro-inflammatory molecules, including cytokines and chemokines [31]. During the hyperglycemic status, the activation of NF-κB induces pro-inflammatory mediators, such as TNF-α.

The results of the current study show the ability of *X. testudinaria* extract to reverse the abnormal elevation of NF-κB. This effect well explained the effect of the *X. testudinaria* extract on the suppression of TNF-α. It explains the neuroprotective effect of *X. testudinaria* extract in line with the current literature, which shows the effectiveness of anti-inflammatory agents in the management of diabetic peripheral neuropathy [32].

Hyperglycemia contributes to the development of diabetic peripheral neuropathy by triggering oxidative stress and inflammation that leads eventually to neuronal dysfunction, as shown in Figure 9. *X. testudinaria* extract could improve diabetic patient hyperalgesia through various antidiabetic-independent mechanisms. *X. testudinaria* extract improves the neuronal oxidative status not only by rising the endogenous antioxidant, but also by suppressing the ROS. Additionally, it suppresses the pro-inflammatory mediators, such as TNF-α, by inhibiting NF-κB.

## 4. Materials and Methods

### 4.1. Materials

#### 4.1.1. *Xestospongia Testudinaria*

The sponge was collected by hand using divers on Ghurab Reef in the Red Sea at Jazan, Saudi Arabia (15–30 m depth, on May 2019). The sponge was frozen immediately following the collection and freeze-dried. The sponge was identified as *X. testudinaria* by Prof. Rob van Soest at the Naturalis Biodiversity Center at Leiden in the Netherlands. A sample of the sponge is reserved in the collections of the Naturalis Biodiversity Center (RMNH Por. 9176). Another specimen was placed in the Red Sea Marine Invertebrates Collection, at the Faculty of Pharmacy, King Abdulaziz University (No. DY-KSA-12). A complete description of the isolated compounds from the sponge (*X. testudinaria*) and their NMR spectral data have previously been published [11,12].

#### 4.1.2. Animals

Male mice (25–30 g) were bought and housed at the animal facility of the Faculty of Pharmacy, King Abdulaziz University, in Jeddah, Saudi Arabia. Animals were segregated into 3 animals per cage. The animals were allowed free access to food and water on a 12/12 light–dark cycle. During the experiment, the animal housing temperature was maintained at 23 ± 1 °C with a relative humidity of 55 ± 10%.

### 4.2. Ethical Approval

Animal handling and all in vitro and in vivo procedures were performed according to ethical guidelines and approved by the ethics committee of the Faculty of Medicine at King Abdulaziz University (No. 237-18). All efforts were taken to reduce the quantity and degree of suffering in the experimental animals.

### 4.3. Methods

#### 4.3.1. Extraction of the Sponge and Preparation of the Extract Was Carried Out as Follows

The freeze-dried sponge material (150 g) was extracted with methanol (500 mL × 3) at room temperature. The combined methanolic extracts were dried under a vacuum. The resulting extract was dissolved in water and freeze-dried. The freeze-dried extract was given using an IP injection in the experimental mice, as described below.

#### 4.3.2. DRG Isolation and Culture

The detailed method of the mice dorsal root ganglia (DRG) isolation and culture has previously been published [15]. In brief, the collected DRGs of the mice underwent an enzymatic digestion by incubating the DRGs in a mixture of 0.06 μg/mL of collagenase XI (Sigma, St. Louis, MI, USA) and 0.1 μg/mL of dispase (Sigma) for 1 h at 37 °C and 5% CO_2_. After washing, the cells were resuspended in neurobasal A media (NBA, Gibco, Waltham, MA, USA), which contains 25 mM of glucose with a 2 mM Glutamax supplement (Gibco), 1% penicillin/streptomycin (Gibco), 2% B-27 supplement (Gibco), and 10 ng/mL of nerve growth factor (Sigma). Following this, the cells were seeded on Poly-D-Lysine/laminin (BD)-coated plates.

#### 4.3.3. In Vitro and in Vivo Induction of Diabetic Peripheral Neuropathy Model

In vitro induction:

To induce an in vitro model of diabetic peripheral neuropathy, some of the DRG cultures were incubated with 45 mM of high glucose media for 24 h at 37 °C and 5% CO_2_, as reported in previous published research [33].

In vivo induction:

The detailed method has previously been published [34]. The animals were considered as diabetic when their fasting blood sugar exceeded 250 mg/dL, as measured with a glucometer (AccuChekperforma, Roche, Switzerland).

#### 4.3.4. In Vitro and In Vivo Experimental Design

In vitro design:

Before starting the in vitro experiment, the safety of the different concentrations (0.25, 5, 1, and 2 mg/mL) of *X. testudinaria* has been evaluated on DRG cultures for 24 h. All the tested concentrations showed no effect on the neuronal viability. Therefore, the concentrations of 1 and 2 mg/mL of *X. testudinaria* have been selected to test the effect of *X. testudinaria* on the HG media-induced DM in vitro model.

The DRG cultures were grouped into four groups: the first group (the control group) was given a standard glucose concentration (25 mM of glucose), which was optimum for neuron growth. The second group was the high glucose (HG) group (45 mM of glucose). The third group, the 2 mg/mL of sponge extract-treated group, was given to a high glucose (45 mM of glucose) media. The fourth group, the 100 μM of α-lipoic acid-treated group, was given a high glucose (45 mM) as a reference group because it is well known as an antioxidant and has a potent antioxidant activity in ROS scavenging [35]. The dose of 2 mg/mL of sponge extract was chosen based on the results from the experiments shown in Figure 1.

In vivo design:

Twenty-four male mice (25–30 g) were randomly divided into four groups (6 mice each):

Group 1 (control): received only the vehicle (normal saline) IP for 35 days.

Group 2 (sponge extract group): received 0.75 mg/kg of sponge extract IP for 35 days.

Group 3 (STZ): served as a diabetic mice group and received 150 mg/kg of STZ intraperitoneally once [36], then received normal saline IP for 35 days.

Group 4 (STZ + sponge extract group): diabetic mice group received 150 mg/kg of STZ intraperitoneally once, then treated with 0.75 mg/kg/d IP sponge extract for 35 days. The intraperitoneal route of administration of the *X. testudinaria* was selected because the pharmacokinetic of the *X. testudinaria* is unknown and there is no previous study regarding the pharmacokinetics of the *X. testudinaria* in human or animals. Therefore, an IP administration of *X. testudinaria* was chosen as the route of administration to avoid the risk of food–drug interaction and other factors that may influence the bioavailability of *X. testudinaria.*

#### 4.3.5. In vitro Studies to Evaluate the Neuroprotective Effect of Sponge Extract Using Viability Assay

The cells viability was assessed using a calcine-AM (non-fluorescent lipophilic dye; Invitrogen, Life Technologies, Waltham, MA, USA) fluorescent assay kit. The DRG cells were cultured with or without the sponge extract for 24 h. The next day, 5 μm of calcine was added to each well and incubated in the dark at 37 °C for 45 min. Following this, the fluorescence of the viable cells was measured with a fluorescent microplate reader (BioTek^®^, Synergy HT, Winooski, VT, USA) at excitation and emission wavelengths of 480 and 520 nm, respectively.

#### 4.3.6. Immunocytochemistry

After washing and fixing the cells with 4% of paraformaldehyde for 10 min, Triton X100 was added for 10 min to enhance the cell permeability. The cells were incubated with a blocking buffer (containing 2% goat serum (Sigma), 0.2% fish serum gelatine (Sigma), and 0.025% Triton X100 in PBS) for 1 h. The cells were then stained with mouse anti-b III Tubulin monoclonal IgG (R&D MAB1195 clone TuJ-1 lot HGQ0113121, 1:1000) at 4 °C overnight. The next day, the cells were washed with PBS. The cells were stained with the Alexa Fluor anti mouse secondary antibodies (1:1000, Invitrogen) and kept in the dark at room temperature for 2 h. Following this, a drop of 4′, 6-diamidino-2-phenylindole (DAPI; Vectashield Hard set H1500, Vector) was used to stain the nuclei. A duplicate coverslip was made for each treatment condition. The experiment was carried out 3 times. A fluorescent microscope was used to capture the image at 20× and 40× objectives. For each coverslip, 4 different pictures were taken, and image analysis was carried out using Fiji-Image J (Schneider, 2012) by computing the neurite length. The neurite length in each section was measured by quantifying the ratio of the total number of neurite lengths (in μm) over that of the number of neuronal cell bodies [37].

#### 4.3.7. In Vivo Studies to Evaluate the Neuroprotective Effect of Sponge Extract by Evaluating Hot Hyperalgesia

Diabetic peripheral neuropathy was measured by hot hyperalgesia using hot plate analgesiometer at the beginning and end of the experiment, with an analgesiometer (model 35100; Basile, Italy). After adjusting the plate to 48 °C, the mice were placed on the plate, and the time from the initial heat exposure to the withdrawal of the hind paw was measured (the latency period). The maximum time was set as 30 s to protect animal tissue injury.

#### 4.3.8. Evaluation of Oxidative Stress and Pro-Inflammatory Biomarkers

The lysate from DRG was prepared by incubating the DRGs with 1% of protease inhibitor cocktail III (Fisher Scientific, St. Louis, MI, USA) for 1 h. The protein contents in the supernatant were quantified using a Bradford protein assay.

The oxidative stress parameters were measured using mice malondialdehyde (MDA) ELISA kit (MyBioSource MBS263626, San Diego, CA, USA), a mice glutathione (GSH) ELISA kit (MyBioSource MBS267424, San Diego, CA, USA), and a mice superoxide dismutase enzymes (SOD) ELISA kit (MyBioSource MBS034842, San Diego, CA, USA), following the manufacturer’s instructions.

The pro-inflammatory biomarkers were measured using a mouse nuclear factor KB ELISA Kit (MyBioSource MBS043224, San Diego, CA, USA) and the tumor necrosis factor kit assay (MyBioSource MBS825075, San Diego, CA, USA), following the manufacturer’s instructions.

#### 4.3.9. Histological Examination

At the end of the experiment, the following specimens were collected and processed for a light microscopic examination. The liver specimens were taken from the right lobe. The pancreatic specimens were collected from the tail of the pancreas. Lumbar DRGs were removed bilaterally. All the collected tissues were then fixed in 10% buffered formalin for 1 week, and then they were dehydrated in ascending grades of alcohol. Then, the specimens were cleared in xylene; 2 changes took place for half an hour each. Following this, the specimens were embedded in 2 changes of soft paraffin, 1 h for each in an oven with a melting point of 55–57 °C. Paraffin blocks were obtained and serial sections of a 5 µm thickness were made by a microtome (Leica RM2035) and mounted on glass slides. The consecutive slides were subjected to the following stains:(1)Hematoxylin and Eosin (H & E) stain for all specimens for a routine histological examination.(2)Masson’s Trichrome stain for the liver and pancreas to show the collagen fibers [17].

Periodic Acid Schiff (PAS) stain to show the glycogen storage in the liver and pancreas [38].

(3)Reticulin stain for DRG demonstration of reticulin fibers [39].

The sections were examined and photographed using a light microscope (model: BX51TF, Japan).

Following the staining, the findings in the liver sections were photographed at a magnification of 10×, and in the pancreatic specimens at a magnification of 20×. For the DRG specimens, the magnification was set at 40×.

### 4.4. Statistical Analysis

All data and graphs were analyzed using GraphPad Prism, version 9 (GraphPad Software, La Jolla, CA, USA). Data were expressed as mean +/− standard error of mean (SEM) or as mean +/− standard deviation (SD), as parametric data followed a normal distribution. All statistical comparisons were made for independent parametric parameters using a one-way ANOVA followed by Tukey post hoc test. A *p* value of <0.05 was considered statistically significant. For all statistical comparisons, Bartlett’s test was used for the assessment of data homogeneity and normality.

## 5. Conclusions

The current study was carried out to evaluate the safety and the effectiveness of *X. testudinaria* extract in diabetic peripheral neuropathy. The results provide evidence for the neuroprotective effect of *X. testudinaria* extract, using in vitro and in vivo models of diabetic peripheral neuropathy. This might be a product of its antioxidant and anti-inflammatory mechanisms. However, further work is required to investigate the effects on the signaling pathway. In particular, human clinical trials are needed to evaluate further the efficacy and safety of *X. testudinaria* extract on diabetic peripheral neuropathy.

## Figures and Tables

**Figure 1 pharmaceuticals-15-01309-f001:**
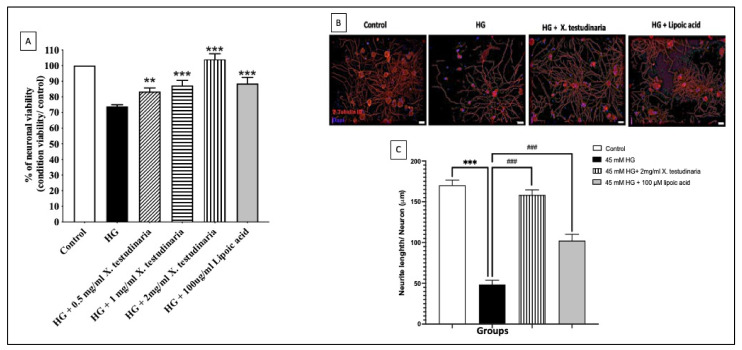
The effect of *X. testudinaria* extract on DRG neurons. (**A**) shows the effect of *X. testudinaria* extract on DRG viability. (**B**) shows the schematic tracings of the neurites from DRG culture in different groups. Neurites and nuclei were visualized with β-tubulin III and DAPI, respectively. (**C**) shows a quantitative analysis of total neurites length in DRG cultures. Data are presented as mean ± SEM of three independent experiments. Statistical analysis was carried out using a one-way ANOVA followed by Tukey post hoc test. **, ***: the data were statistically significant from the corresponding control group at *p* < 0.01 and *p* < 0.001, respectively. ###: the data were statistically significant from the corresponding high glucose (HG) group at *p* < 0.001.

**Figure 2 pharmaceuticals-15-01309-f002:**
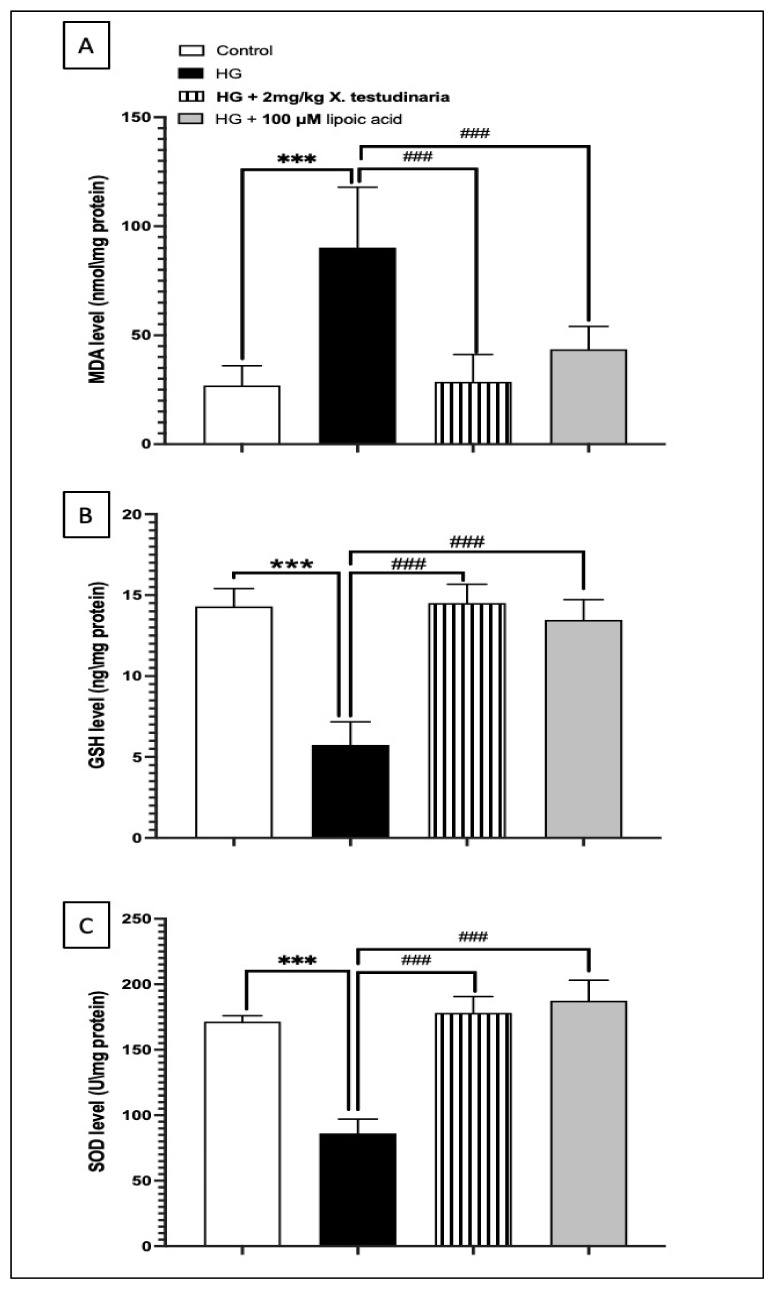
The effect of *X. testudinaria* extract on oxidative stress biomarkers in high glucose (HG) -DRG culture: (**A**) malondialdehyde (MDA) level, (**B**) glutathione (GSH), and (**C**) superoxide dismutase enzymes (SOD). Data are presented as mean ± SEM for the three independent experiments. Statistical analysis was carried out using a one-way ANOVA followed by Tukey post hoc test. ***: statistical significance from the corresponding control group was set at *p* < 0.001. ###: statistical significance from the corresponding HG group was set at *p* < 0.001.

**Figure 3 pharmaceuticals-15-01309-f003:**
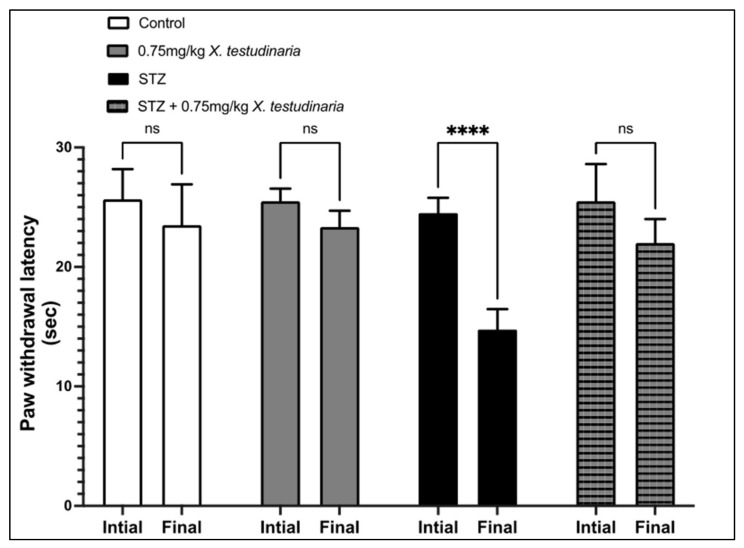
Paw withdrawal latency to hot stimuli in streptozotocin (STZ)-induced diabetic mice. Data are presented as mean ± SD for the mice (n = 6) in each group. Statistical analysis was carried out using one-way ANOVA followed by Tukey post hoc test. ****: statistical significance from the corresponding initial reading was set at *p* < 0.001. ns: nonsignificant.

**Figure 4 pharmaceuticals-15-01309-f004:**
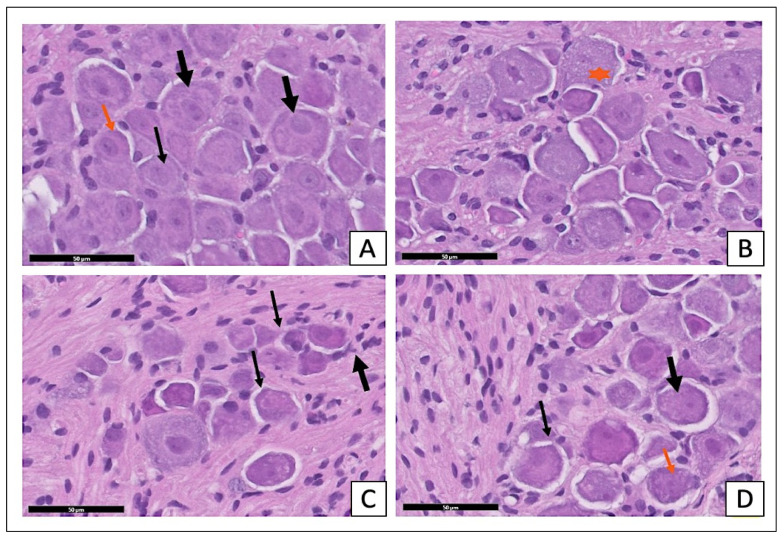
Effect of *X. testudinaria* extract on histopathological changes of DRG in streptozotocin (STZ)-induced diabetes in mice using H and E: a photomicrograph of the dorsal root ganglia (DRG) of the control group (**A**), *X. testudinaria* extract-only group (**B**), STZ-diabetic group (**C**), and STZ-diabetic on *X. testudinaria*-treated group (**D**). Arrows show neurons and stars mark large-diameter nerve cells. Thick black arrows show satellite cells. H&E scale bar at 50 µm was used.

**Figure 5 pharmaceuticals-15-01309-f005:**
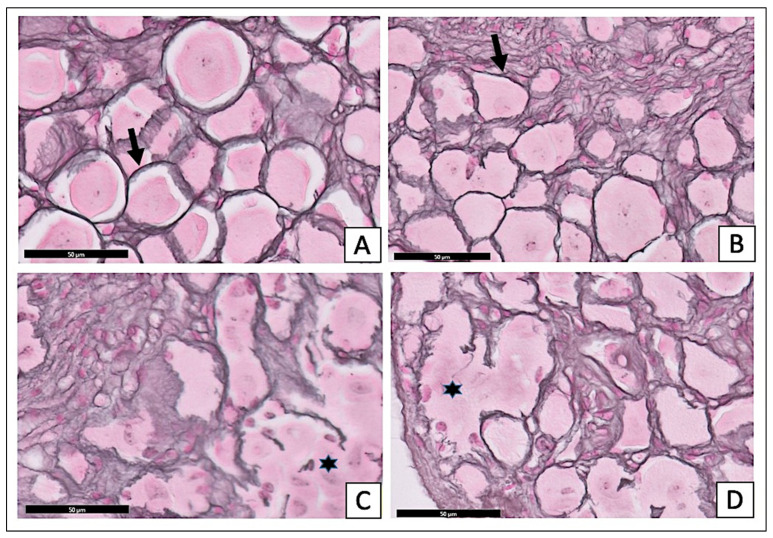
Effect of *X. testudinaria* extract on histopathological changes of DRG in streptozotocin (STZ)-induced diabetes in mice using reticulin stain: a photomicrograph of the dorsal root ganglia (DRG) of control group (**A**), *X. testudinaria*-only group (**B**), STZ-diabetic group (**C**), and STZ-diabetic on *X. testudinaria*-treated group (**D**). Arrows show reticular stroma and stars mark the distorted stroma, which give rise to the alveolar pattern. Reticulin stain was used and a scale bar 50 µm.

**Figure 6 pharmaceuticals-15-01309-f006:**
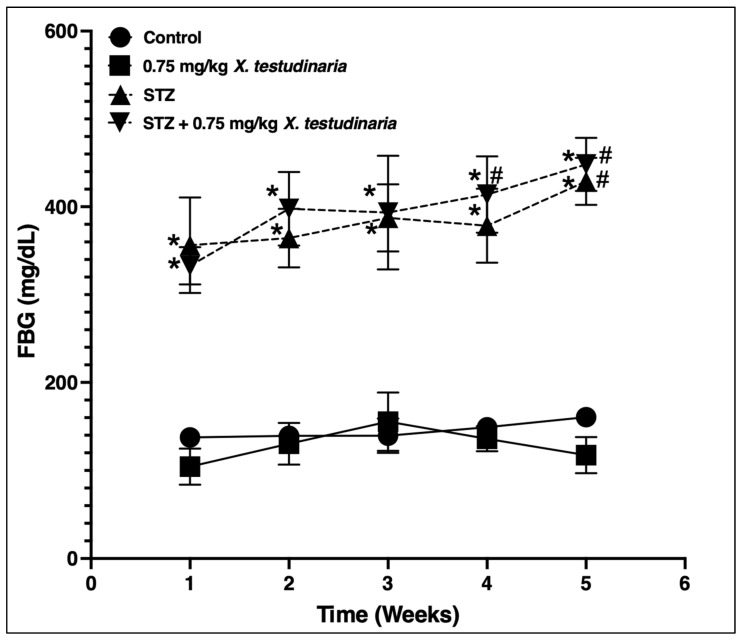
Fasting blood glucose (FBG) level in streptozotocin (STZ)-induced diabetic mice. Data are presented as mean ± SD for the mice (n = 6) in each group. Statistical analysis was carried out using a one-way ANOVA followed by Tukey post hoc test. *: statistical significance from the corresponding control group was set at *p* < 0.05. #: statistical significance from the corresponding week 0 was set at *p* < 0.05.

**Figure 7 pharmaceuticals-15-01309-f007:**
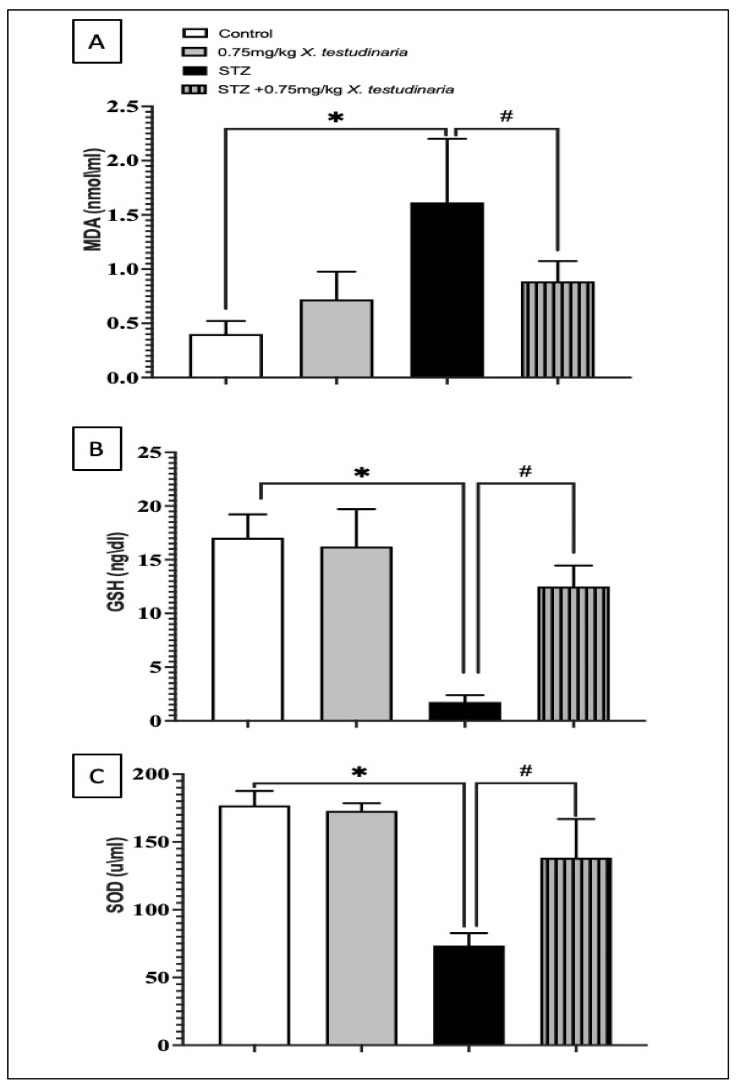
Effect of *X. testudinaria* extract on oxidative biomarkers in the dorsal root ganglia (DRG) of streptozotocin (STZ)-induced diabetic neuropathic mice: (**A**) malondialdehyde (MDA) level, (**B**) glutathione (GSH) level, and (**C**) superoxide dismutase enzymes (SOD) level. Data are presented as mean ± SD for mice (n = 6) in each group. Statistical analysis was carried out using a one-way ANOVA followed by Tukey post hoc test. *: statistical significance from the corresponding control group was set at *p* < 0.05. #: statistical significance from the corresponding STZ group was set at *p* < 0.05.

**Figure 8 pharmaceuticals-15-01309-f008:**
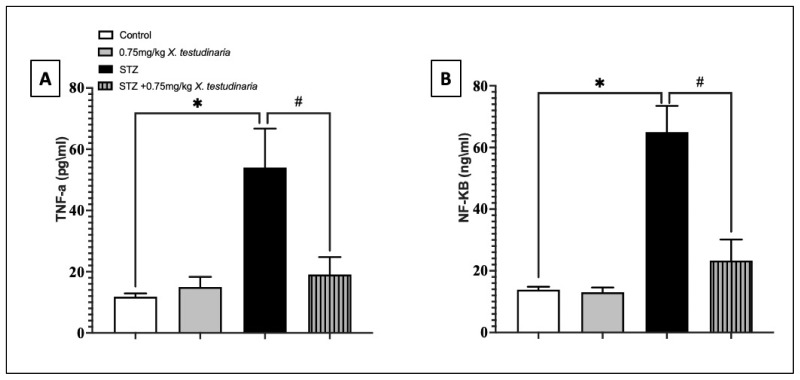
Effect of *X. testudinaria* extract on tumor necrosis factor (TNF)-α (**A**) and nuclear factor kappa B (NF-κB) (**B**) content in DRG of streptozotocin (STZ)-induced diabetic neuropathic mice. Data are presented as mean ± SD for the mice (n = 6) in each group. Statistical analysis was carried out using a one-way ANOVA followed by Tukey post hoc test. *: statistical significance from the corresponding control group was set at *p* < 0.05. #: statistically significance from the corresponding STZ group was set at *p* < 0.05.

**Figure 9 pharmaceuticals-15-01309-f009:**
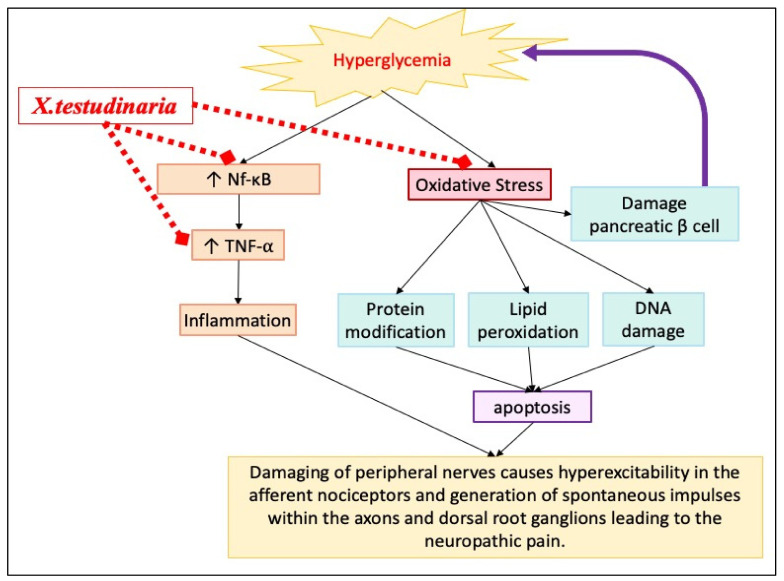
A schematic overview of pathogenesis of diabetic neuropathy and possible mechanism (s) through which *X. testudinaria* extract could exert its diabetic neuroprotective effects.

## Data Availability

Data is contained within the article and Appendix A.

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
