# Peer review of "Neuroprotective Effect of Red Sea Marine Sponge Xestospongia testudinaria Extract Using In Vitro and In Vivo Diabetic Peripheral Neuropathy Models"

_pharmaceuticals, 2022, doi:10.3390/ph15111309_

Round 1
Reviewer 1 Report
The manuscript entitled “Neuroprotective effect of Red Sea Marine Sponge Xestospongia testudinaria extract using in vitro and in vivo diabetic peripheral neuropathy models” addresses the beneficial neuroprotective effects of the Red Sea marine sponge Xestospongia testudinaria extract in high-glucose triggered neuronal viability and neurite length (in vitro model). Moreover, the manuscript assesses the neuroprotective potential in diabetes mellitus-induced neuropathy model in mice (in vivo model) and associated molecular mechanisms. At the molecular levels, the extract dampened some oxidative stress and pro-inflammatory markers. The manuscript is clearly written, and the current findings are interesting.
Comments:
1) For X. testudinaria extract, the authors are advised to add all the chemical charts (e.g., HPLC, NMR, ….) that confirm the identity of isolated compounds as a supplementary file. This is essential to verify the reliability of the present results.
2) In the experimental protocol section, how did the authors decide on the dose of X. testudinaria extract (0.75 mg/kg) in mice? How can the dose How is the dose relevant to the human dose using the Human effective dose (HED) formula= animal dose x animal Km/ human Km (Nair AB, Jacob S. A simple practice guide for dose conversion between animals and humans. J Basic Clin Pharm. 2016 Mar;7(2):27-31). Authors are advised to address this point and add the answers to the comment in the experimental protocol section. Please also provide proper citations for selecting this dose.
3) Likewise, how did the authors decide on the dose of X. testudinaria extract (2 mg/mL) in the in vitro studies? Authors are advised to address this point and add the answers to the comment in the experimental protocol section. Please also provide proper citations for selecting this dose.
4) In the introduction section, the authors are advised to describe whether the major constituents of the extract can cross the nervous tissue in order to elicit these neuroprotective actions. Or did any previous literature address this point?
5) In the statistical analysis section, did the authors check data normality and homogeneity before proceeding to one-way ANOVA? Authors are advised to address this point and add the answers to the comment in the material and methods section.
6) In line 399, why did the authors choose the intraperitoneal route for the administration of the extract? In fact, it is more common that humans administer extracts via the oral route. Kindly, discuss the rationale, and please also provide the rationale for the timing and frequency of administration. Authors are advised to address this point and add the answers to the comment in the material and methods section.
7) In the statistics section, the name of the post-hoc test used in the statistical analysis should be clearly stated.
8) To make all figure legends stand-alone, authors are advised to add the full name of the abbreviations at each legend's end. Moreover, the name of the post-hoc test used in the statistical analysis should be clearly stated.
9) the statistical comparisons in figure 3 are not proper since they only show the difference between initial and final measurements for each group. However, the proper comparison is the one that compares whether significant changes exist between STZ gp vs STZ + extract. Please, address this issue.
10) the authors are advised to adjust the figures properly in the manuscript since part of each figure is cut and not displayed properly.
11) The authors are advised not to overestimate the obtained results. For example, in lines 449-450, the authors state that “The results provide substantial evidence for the neuroprotective effect of X. testudinaria extract using in vivo and in vivo models of diabetic peripheral neuropathy”. Please, remove “substantially”.
12) In line 360, please correct “In vivo design” to “In vitro design”.
13) More recent 2021-2022 references are advised to be added to the current manuscript.
Author Response
Dear Editor-in-Chief:
We thank you and the reviewers for the valuable and constructive comments. We have carefully reviewed the comments, and the manuscript was revised according to the suggestions provided by you. Our detailed responses are as follows. We have marked the modifications we made in red. The revised version of the manuscript has been approved by all authors.
We hope that the revised version is suitable for publication in Pharmaceuticals. We look forward to hearing from you, and we would be happy to respond to any other question or comment that you may have.
Please find the file attached.
Sincerely,
Corresponding author
AZIZA RASHED ALRAFIAH

Reviewer 2 Report
The chemical composition and isolation methods of the methanol/ethanol extracts of the Red Sea Marine Sponge Xestospongia testudinaria are already reported in the literature (references 10-14 from the manuscript).
It is described that some components of the sponge extract possess cytotoxic activity. Therefore the methanolic extract tested by the authors could contain these components that could exert this cytotoxic activity on the cell cultures they tested.
This fact could almost certainly have influenced the results they had obtained in cell culture tests. To exclude/evaluate this latter possibility, the authors have to carry out cytotoxicity tests of the sponge extract on the cells they used in the cultures presented in the study.
In addition, various fraction extracts more or less rich in one or more chemical components have been described (references 10-14 from the manuscript). Authors need to explain in the introduction and/or discussion sections why they tested a methanol extract, that contains a large mix of chemical components, and not fractions containing the individual chemical component as previously identified. If they had proceeded latterly, it would have been possible to obtain very detailed information on the activity of the individual components of the extract which would have generated very valuable information useful, i.e., for medicinal chemistry design, or the formulation of new nutraceuticals.
The authors state that the sponge extract is more active than Lipoic acid tested at a concentration of 100 micromoles/ml. The authors have to specify in the discussion on which bases they chose this concentration of lipoic acid before affirming that it is less active than their methanolic extract.
A bibliographic search conducted on Google Scholar resulted in a publication that deals with topics that are identical to those that the authors propose in this manuscript (please see below and attached files*). Furthermore, one of the authors (Kariman Jamal Borouk) appears to be involved in this publication. Unfortunately, we were unable to download this article because the website (JS ARABIA - platform.almanhal.com) does not allow it. Therefore, the authors must provide this article and mention it in the introduction and the list of references.
The whole manuscript needs to be revised to eliminate various imperfections, for example, the companies that supplied the MDA, GSH, and SOD assay kits are not specified
*Please follow the link reported below or search on Google Scholar:
Red Sea Marine Sponge Xestospongia 2 testudinaria "diabetic peripheral neuropathy"
https://scholar.google.com/scholar?hl=en&as_sdt=0%2C5&q=Red+Sea+Marine+Sponge+Xestospongia+2+testudinaria+%22diabetic+peripheral+neuropathy%22++&btnG=]

Author Response

(The authors gave the same response as above.)

Round 2
Reviewer 2 Report
Dear authors thanks for your elucidations. The manuscript now is ok.